# Modelling and Comparative Analysis of Different Methods of Liquid Membrane Separations

**DOI:** 10.3390/membranes13060554

**Published:** 2023-05-26

**Authors:** Artak E. Kostanyan, Andrey A. Voshkin, Vera V. Belova, Yulia A. Zakhodyaeva

**Affiliations:** Kurnakov Institute of General and Inorganic Chemistry, Russian Academy of Sciences, 31 Leninskii pr., 119991 Moscow, Russia; voshkin@igic.ras.ru (A.A.V.); belova@igic.ras.ru (V.V.B.); yz.igic@gmail.com (Y.A.Z.)

**Keywords:** extraction-stripping separation processes, emulsion and supported liquid membranes, film pertraction, three-phase extraction, mathematical modelling

## Abstract

This article is devoted to a brief review of the modelling of liquid membrane separation methods, such as emulsion, supported liquid membranes, film pertraction, and three-phase and multi-phase extraction. Mathematical models and comparative analyses of liquid membrane separations with different flow modes of contacting liquid phases are presented. A comparison of the processes of conventional and liquid membrane separations is carried out under the following assumptions: mass transfer is described by the traditional mass transfer equation; the equilibrium distribution coefficients of a component passing from one of the phases to another are constant. It is shown that, from the point of view of mass transfer driving forces, emulsion and film pertraction liquid membrane methods have advantages over the conventional conjugated extraction stripping method, when the mass-transfer efficiency of the extraction stage is significantly higher than the efficiency of the stripping stage. The comparison of the supported liquid membrane with conjugated extraction stripping showed that when mass-transfer rates on the extraction and stripping sides are different, the liquid membrane method is more efficient, while when they are equal to each other, both processes demonstrate the same results. The advantages and disadvantages of liquid membrane methods are discussed. The main disadvantages of liquid membrane methods—low throughput and complexity—can be overcome by using modified solvent extraction equipment to carry out liquid membrane separations.

## 1. Introduction

There are practical situations when, during mass transfer or heat transfer processes, direct contact of two interacting phases is undesirable or impossible due, for example, to their solubility or low selectivity in the mass transfer and thermal stability of one of them—in heat transfer. Solid–solid, liquid–liquid, or solid–liquid pairs can be considered as interacting phases. To carry out transfer processes in such situations, a transfer medium can be used that contacts both phases simultaneously or alternately, and performs heat or mass transfer between them. The transfer medium may be stationary in the operating system or may circulate in the system, with the first and second phases to be brought into the interaction, eluting through it as the mobile phases. In this article, as an example of non-contact transfer processes, mass transfer processes between two miscible liquids, namely, sequential (conjugate) extraction–back extraction and liquid membranes, are considered. Liquid membrane extraction techniques are a promising alternative to conventional conjugated extraction-stripping separation processes. Compared to solvent extraction (SE) methods, liquid membrane (LM) methods have a higher potential for extracting, purifying, and enriching a wide range of pharmaceuticals and chemicals (including radionuclides) from dilute aqueous solutions, avoiding the disadvantages of SE, such as the high consumption of solvents. There are a number of different methods of liquid membrane separation. These techniques are based on a three-phase system with an organic phase held stationary in the operating system (or it circulates in the system) while the first (feed) and second (stripping) aqueous phases are eluted through it as the mobile phases. It should be noted that the processes of extraction-stripping separation can be carried out both in continuous and discontinuous (staged) versions. Figure 1 shows schematic diagrams of continuous versions of a conventional conjugated extraction-stripping process (a), a film pertraction process (b), and emulsion membrane (c) and supported liquid membrane (d) separation processes. 

The technique of liquid film pertraction has been shown to provide a simple and stable operation with high recovery rates [1]. The possibility of the selective separation of metals, mineral, and organic acids, or gases dissolved in water has been established. This method can also be used to separate biologically active substances, enzymes, natural and synthetic drugs, etc. [2]. However, due to low throughput, this method is hardly applicable in large-scale separation processes, in particular, in hydrometallurgy. The conventional extraction-stripping process includes an extraction column and a stripping column; in the film pertraction, three phases flow as parallel films [1,2,3]. In the emulsion membrane extraction, two options are possible: (1) the aqueous stripping phase is encapsulated as microdroplets in large droplets of a liquid membrane moving in a continuous aqueous feed solution [4,5,6,7,8,9,10,11,12,13,14]; and (2) the aqueous feed phase is encapsulated as microdroplets in large droplets of the liquid membrane phase rising or falling in the continuous stripping phase. Emulsion liquid membranes are double emulsions because they consist of water–oil–water or oil–water–oil systems. In the first case, an emulsion liquid membrane system is obtained by combining two immiscible phases and then mixing the resulting emulsion with an external aqueous phase to form a water–oil–water emulsion [6]. Surfactants are often used to stabilize emulsions by micelle formation. Emulsion membranes have some advantages over the extraction method, such as easy operation, suitability for low and high concentrations of metal ions, simultaneous extraction and distillation, low power consumption, and low cost: the liquid membrane, which includes both the carrier and the surfactant, can be reused at the end of the process [5]. However, due to low productivity, this method also cannot be used in large-scale separation processes.

In the supported liquid membranes, the organic liquid membrane is sandwiched between the aqueous feed and stripping solutions, which move on opposite sides of a porous polymer impregnated with the organic liquid membrane [15,16,17,18,19,20,21,22,23,24]. In all cases, the mass transfer occurs between the two aqueous phases through the immiscible organic membrane phase.

Membrane tubes are often used as mass transfer devices, as well as flat membranes made of polypropylene, polyvinylidene fluoride, polytetrafluoroethylene, and polysulfone. The first membrane contactors were made from tubular polypropylene; however, the chemical and thermal stability of such membranes was poor. Recently, membrane contactors with tubular membranes produced from polytetraflourethylene have been offered, which are highly resistant to aggressive media. For the treatment of liquids with the production of drinking water, for large water utilities, ultrapure water for energy and medicine, and sea water desalination, ultrafiltration hollow fiber membranes with a spongy and finger-like pore structure have been developed, manufactured, and tested [16]. Supported liquid membranes are widely used to separate metals from model and industrial wastewater [15,17,20,21,22,23,24]. The stability of the long-term operation of these membranes is affected by the chemical and technical characteristics of the polymer substrate, the nature of the organic phase of the membrane, and the method of preparation of the membranes. Flat sheet membranes, which are used in sandwich modules, are impregnated by immersion in an organic phase containing a carrier, but, in this case, there is no control over the penetration of the carrier into the pores of the polymer substrate. Therefore, in most systems with such membranes, after a long time of operation, a decrease in the flows of the aqueous phase is observed. The poor stability of flat sheet membranes limits their long-term use, so polymer membranes have been developed. In the field of membrane separation, of particular interest are solid membranes for molecular separation processes [25]. Studies of macrocyclic hosts based on MXene have shown their great potential for development and wide application in membrane catalysis and membrane separation [26].

To select an appropriate method for solving the set separation problem, as well as for its optimal design, preliminary mathematical modelling is necessary. A brief overview of the mathematical modelling of the separation processes under consideration, carried out by the authors [27,28,29,30,31,32,33], is presented below.

## 2. Modelling of Continuous Versions of Extraction-Stripping Separation

In Figure 2, the flowsheet of mass transfer in a conventional conjugated extraction-stripping process is shown. In Figure 3, the possible schemes of phase flows in the liquid membrane technique are shown: (a) film pertraction—all three phases (feed or raffinate, liquid membrane, and extract or strip) flow as parallel films; emulsion membranes—three schemes of phase flows are possible (a–c); and supported liquid membranes (d,e).

According to Figure 2 and Figure 3, the mathematical models of the corresponding extraction-stripping processes for conditions of constant equilibrium distribution coefficients can be represented by the following equations of material balance and mass transfer:

Conventional extraction-stripping process (Figure 2):(1)$$-v_{1}\frac{dx_{1}}{dz}=a_{1}k_{1}(x_{1}-y_{1}^{*}/m_{1})=-w\frac{dy_{1}}{dz}$$
(2)$$-v_{2}\frac{dx_{2}}{dz}=a_{2}k_{2}(y_{2}^{*}-m_{2}x_{2}^{*})=-w\frac{dy_{2}}{dz}$$
(3)$$v_{1}x_{1}+wy_{L}=v_{1}x_{1,L}+wy_{1}$$
(4)$$wy_{2}=v_{2}x_{2}+wy_{L}$$

Equations (1) and (2) describe the processes of extraction and back-extraction (stripping) in the extraction and stripping columns, respectively. In Equations (1)–(4), $x_{1}$ is the concentration of the passing component in the aqueous feed phase (raffinate or donor phase) in the extraction column, g/mL or g/cm^3^; $y_{1}$ is the concentration of the passing component in the organic phase (extractant phase) in the extraction column, g/mL or g/cm^3^; $x_{2}$ is the concentration of the passing component in the aqueous stripping phase (acceptor or extract phase) in the stripping column, g/mL or g/cm^3^; $y_{2}$ is the concentration of the passing component in the organic phase in the stripping column, g/mL or g/cm^3^; $a_{1}$ and $a_{2}$ are the specific areas of phase contact in the extraction and stripping columns, respectively, cm^−1^; $k_{1}$ and $k_{2}$ are mass-transfer coefficients in the extraction and stripping columns, respectively, cm/s; $v_{1}$ and $v_{2}$ are specific flow rates of the aqueous phases in the extraction and stripping columns, respectively, cm/s; *w* is the specific flow rate of the organic phase in the extraction and stripping columns (in the extraction column, it is the extract phase; in the stripping column, it is the feed or the raffinate phase), cm/s; $m_{1}=\frac{y^{*}}{x_{1}^{*}}=constant$ and $m_{2}=y^{*}/x_{2}^{*}=constant$ are the equilibrium distribution coefficients in the extraction and stripping columns, respectively; and *z* is the co-ordinate along the phase contact surface, cm (0≤z≤L).

Film pertraction extraction (Figure 3, scheme (Figure 3a)):(5)$$-v_{1}\frac{dx_{1}}{dz}=a_{1}k_{1}(x_{1}-y^{*}/m_{1})$$
(6)$$w\frac{dy}{dz}=a_{1}k_{1}(x_{1}-y^{*}/m_{1})-a_{2}k_{2}(y-m_{2}x_{2}^{*})$$
(7)$$v_{1}x_{1,0}+wy_{0}=v_{1}x_{1}+wy+v_{2}x_{2}$$
(8)$$y_{0}=y_{L}$$

Equations (5) and (6) describe the mass transfer on the extraction and stripping sides of the film pertraction extraction process with co-current flows of feed, liquid membrane, and extract phases.

Emulsion liquid membrane extraction (Figure 3, scheme (Figure 3b,c))

Scheme (Figure 3b): co-current flow of the feed and membrane phases with the countercurrent flow of the extract phase (the aqueous feed solution is encapsulated as microdroplets in organic liquid membrane droplets): 

Equations (5), (6), and (8) remain valid and Equation (7) becomes:(9)$$v_{1}x_{1,0}+wy_{0}+v_{2}x_{2}=v_{1}x_{1}+wy+v_{2}x_{2,0}$$

Scheme (Figure 3c): co-current flow of the extract and membrane phases with the countercurrent flow of the feed phase (the aqueous extract phase is encapsulated as microdroplets in organic liquid membrane droplets):

Equations (5) and (8) remain valid and Equations (6) and (7) become:(10)$$-w\frac{dy}{dz}=a_{1}k_{1}(x_{1}-y^{*}/m_{1})-a_{2}k_{2}(y-m_{2}x_{2}^{*})$$
(11)$$v_{1}x_{1}+wy_{L}=v_{1}x_{1,L}+wy+v_{2}x_{2}$$

Supported liquid membrane extraction

Scheme (Figure 3d): co-current flow of the feed and extract phases:

Equation (5) remains valid:(12)$$a_{1}k_{1}(x_{1}-y^{*}/m_{1})=a_{2}k_{2}(y-m_{2}x_{2}^{*})$$
(13)$$v_{1}x_{1,0}=v_{1}x_{1}+v_{2}x_{2}$$

Scheme (Figure 3e): countercurrent flow of the feed and extract phases:

Equations (5) and (12) remain valid and Equation (13) becomes:(14)$$v_{1}x_{1,0}+v_{2}x_{2}=v_{1}x_{1}+v_{2}x_{2,0}$$

In Equations (5)–(14), $a_{1}$ and $a_{2}$ are the specific areas of phase contact on the extraction and stripping sides of a liquid membrane extraction, respectively; $k_{1}$ and $k_{2}$ are the corresponding mass transfer coefficients; $v_{1}$ and $v_{2}$ are specific flow rates of the aqueous feed and extract phases on the extraction and stripping sides of a liquid membrane extraction, respectively; *w* is the specific flow rate of the phase of the extraction agent (the organic liquid membrane phase); $x_{1}$**,**
$x_{2}$, and *y* are the concentrations of the passing component in the donor and acceptor phases and in the liquid membrane, respectively, and the symbol * stands for equilibrium conditions; and *z* is the co-ordinate over the phase contact surface (0≤z≤L). As mentioned above, the equilibrium distribution coefficients are assumed to be constant ($m_{1}=y^{*}/x_{1}^{*}$ and $m_{2}=y^{*}/x_{2}^{*}$). The boundary conditions for Equations (1)–(17) are shown in Figure 2 and Figure 3. By solving these equations, the outlet concentration in the raffinate can be established as follows:

Extraction-stripping process (Figure 2h):(15)$$\frac{x_{1,L}}{x_{1,0}}=1-\frac{S_{1}}{1+F_{1}S_{1}(1-S_{2})/S_{2}},\;S_{1}=\frac{1-\exp[T_{1}(F_{1}-1)]}{1-F_{1}\exp[T_{1}(F_{1}-1)]},\;S_{2}=\frac{1-\exp[T_{2}(F_{2}-1)]}{1-F_{2}\exp[T_{2}(F_{2}-1)]}$$
where $F_{1}=v_{1}/wm_{1}$ and $F_{2}=wm_{2}/v_{2}$ are the mass-transfer factors in the extraction and stripping columns, respectively (dimensionless parameters); and $T_{1}=a_{1}k_{1}L/v_{1}$ and $T_{2}=a_{2}k_{2}L/w$ are the numbers of transfer units in the extraction and stripping columns, respectively (dimensionless parameters).

Liquid membrane extraction (Figure 3)

Scheme (Figure 3a), film pertraction extraction:(16)$$\frac{x_{1,L}}{x_{1,0}}=1-\frac{S}{A(1+F_{2}+F_{1}F_{2})-F_{2}}$$
where $S=aT_{1}+A(1-\exp r_{1}+ar_{1})$, $A=\frac{1-aT_{1}-b}{1-\exp r_{1}(1+r_{1}/T_{1})+ar_{1}+br_{1}/T_{1}}$, $a=\frac{\exp r_{1}-\exp r_{2}}{r_{1}-r_{2}}$, $b=\frac{r_{1}\exp r_{1}-r_{2}\exp r_{2}}{r_{1}-r_{2}}$, and $r_{1,2}=-0.5\cdot [T_{1}(1+F_{1})+T_{2}(1+F_{2})]\pm \sqrt{0.25\cdot {\left[T_{1}(1+F_{1})-T_{2}(1+F_{2})\right]}^{2}+T_{1}T_{2}F_{1}}$

Emulsion liquid membrane extraction (Figure 3, scheme (Figure 3b,c))

Scheme (b): co-current flow of the feed and membrane phases with the countercurrent flow of the extract phase:(17)$$\frac{x_{1,L}}{x_{1,0}}=1-S/\left[A(1-F_{1}-F_{1}F_{2})+AF_{2}\left\{\frac{a(1+F_{1})T_{1}+b}{A}+\frac{b.r_{1}}{T_{1}}+(a.r_{1}+1)(1+F_{1})-\frac{r_{1}\exp r_{1}}{T_{1}}-(1+F_{1})\exp r_{1}\right\}\right]$$
where $r_{1,2}=0.5\cdot [T_{2}(F_{2}-1)-T_{1}(1+F_{1})]\pm \sqrt{0.25\cdot [T_{1}(1+F_{1})-T_{2}(1-F_{2}){]}^{2}+T_{1}T_{2}F_{1}}$.

The values of *A*, *a*, *b*, and *S* are determined as for Equation (19).

Scheme (Figure 3c): co-current flow of the extract and membrane phases with the countercurrent flow of the feed phase
(18)$$\frac{x_{1,L}}{x_{1,0}}=\frac{(1+r_{1}/T_{1}-b\exp r_{1})\frac{1+d}{1+c}+(a-r_{1}/T_{1})\exp r_{1}}{1+a+F_{1}d-(b-F_{1}c)\frac{1+d}{1+c}}$$
where $a=\frac{r_{2}\exp r_{2}-r_{1}\exp r_{1}}{T_{1}(\exp r_{2}-\exp r_{1})}$, $b=\frac{r_{2}-r_{1}}{T_{1}(\exp r_{2}-\exp r_{1})}$, $c=\frac{F_{2}\left[r_{1}/T_{1}+(1-\exp r_{1})b\right]}{F_{1}F_{2}-F_{2}-1}$, $d=\frac{F_{2}\left[(a-r_{1}/T_{1})\exp r_{1}-a\right]}{F_{1}F_{2}-F_{2}-1}$, and $r_{1,2}=0.5\cdot [T_{2}(F_{2}+1)-T_{1}(1-F_{1})]\pm \sqrt{0.25\cdot [T_{1}(1-F_{1})+T_{2}(1+F_{2}){]}^{2}+T_{1}T_{2}F_{1}}$.

In Equations (16)–(18), $F_{1}=v_{1}/wm_{1}$ and $F_{2}=wm_{2}/v_{2}$ are the mass-transfer factors at the stages of extraction and stripping, respectively (dimensionless parameters); and $T_{1}=a_{1}k_{1}L/v_{1}$ and $T_{2}=a_{2}k_{2}L/w$ are the numbers of transfer units at the extraction and stripping stages, respectively (dimensionless parameters).

Supported liquid membrane extraction (Figure 3, Scheme (Figure 3d,e))

Scheme (Figure 3d): co-current flow of the feed and extract phases:(19)$$\frac{x_{1,L}}{x_{1,0}}=1-\frac{m_{1}[1-\exp-\{TK(F+m_{1})/(1+Km_{1})\}]}{F+m_{1}}$$

Scheme (Figure 3e): countercurrent flow of the feed and extract phases:(20)$$\frac{x_{1,L}}{x_{1,0}}=1-\frac{1-\exp\left(TK\frac{F-m_{1}}{1+Km_{1}}\right)}{1-\frac{F}{m_{1}}\exp\left(TK\frac{F-m_{1}}{1+Km_{1}}\right)}$$

In Equations (19) and (20): $T=a_{1}k_{1}L/v_{1}$; $K=a_{2}k_{2}/(a_{1}k_{1})$; and $F=v_{1}m_{2}/v_{2}$.

Using Equations (15)–(20), the efficiency of the processes under consideration can be compared. Some results of such a comparison are demonstrated in Figure 4. 

It can be estimated that, in terms of driving forces, the liquid membrane methods offer advantages over conventional conjugated extraction stripping only when the mass-transfer efficiency of extraction step is substantially higher than that of the stripping step. When the efficiencies of both steps are comparable or when the efficiency of stripping is higher, conjugated extraction stripping indicates better results except with the supported liquid membrane (scheme (Figure 3e)). At $T_{1}>>1$ and $T_{2}>>1$, the efficiencies of the film pertraction and multiple emulsion processes (schemes Figure 3 (a–c)) cease to be dependent on the absolute values of mass-transfer rates on the extraction and stripping sides and are determined only by their ratio. Under such conditions, the enhancement of mass transfer at the extraction step (e.g., by intensifying the breakup of droplets of the complex emulsion) in the case of the emulsion membrane (scheme (c)) may decrease the total process efficiency.

The comparison of the supported liquid membrane with the countercurrent flow of both mobile phases (scheme (e)) with conjugated extraction stripping (scheme (h)) shows that, when mass-transfer rates in extraction and stripping are different, the scheme (e) is more efficient, when they are equal to each other—both processes demonstrate the same results. The last statement can be illustrated in the following way: taking into account the relationships $\frac{F}{m_{1}}=\frac{v_{1}m_{2}}{v_{2}m_{1}}=F_{1}F_{2}$, $Km_{1}=\frac{a_{2}k_{2}m_{1}}{a_{1}k_{1}}=\frac{T_{2}}{T_{1}F_{1}}$, $KF=\frac{a_{2}k_{2}v_{1}m_{2}}{a_{1}k_{1}v_{2}}=\frac{T_{2}}{T_{1}F_{1}}$, and $T=\frac{a_{1}k_{1}L}{v_{1}}=T_{1}$, Equation (20) can be transformed into
(21)$$\frac{x_{1,L}}{x_{1,0}}=1-\frac{1-\exp\left[\frac{T_{2}(F_{2}-1/F_{1})}{1+T_{2}/(T_{1}F_{1})}\right]}{1-F_{1}F_{2}\exp\left[\frac{T_{2}(F_{2}-1/F_{1})}{1+T_{2}/(T_{1}F_{1})}\right]}$$

It is easy to see that, for $T_{1}=T_{2}=T$ and $F_{1}=F_{1}=F$, the Equations (15) and (21) become identical and reduce to:$$\frac{x_{1,L}}{x_{1,0}}=1-\frac{1-\exp[T(F-1)]}{1-F^{2}\exp[T(F-1)]}$$

## 3. Modelling of Staged Versions of Extraction-Stripping Separation

To overcome the shortcomings of the above liquid membrane methods, such as low productivity and the complexity of devices based on the solvent extraction equipment (mixer-settlers and multistage extraction columns), staged versions of the liquid membrane have been proposed [30]; some of them are presented in Figure 5.

Assuming that the equilibrium distribution of the passing component is attained in each cell of all stages (each cell represents a theoretical plate), the process efficiency can be determined as follows:

Extraction-stripping process:

Scheme (Figure 5h):(22)$$\frac{x_{1,n}}{x_{1,0}}=1-\frac{1}{\frac{1-F_{1}^{n+1}}{1-F_{1}^{n}}+\frac{F_{1}-F_{1}F_{2}}{1/F_{2}^{n}-1}}$$

Scheme (Figure 5b):(23)$$\frac{x_{1,n}}{x_{1,0}}=\frac{F_{1}f^{n-1}}{(1+F_{1})\left[1-F_{2}f^{n-1}+\frac{F_{2}f(f^{n}-1)}{\left(1+F_{2})(f-1\right)}\right]}$$
$$f=\frac{F_{1}+F_{2}+F_{1}F_{2}}{1+F_{1}}$$

Liquid membrane extraction: staged version of supported liquid membranes

Scheme (Figure 5d):(24)$$\frac{x_{1,n}}{x_{1,0}}=\frac{F_{1}F_{2}+{\left(1+F_{2}+1/F_{1}\right)}^{-n}}{1+F_{1}F_{2}}$$

Scheme (Figure 5e):(25)$$\frac{x_{1,n}}{x_{1,0}}=\frac{1-F_{1}F_{2}}{{\left[\frac{1+F_{1}}{(1+F_{2})F_{1}}\right]}^{n}-F_{1}F_{2}}$$

The results of a comparison of staged versions, presented in Figure 6, are identical to those of continuous schemes.

In Figure 7, the numbers of theoretical stages ensuring equal degrees of extraction in schemes (h) and (e) are compared; it demonstrates the advantages of scheme (e) over scheme (h). As the rate of circulation of the extraction agent between the extraction and stripping steps rises, the extraction efficiency of scheme (h) passes a maximum, whereas that of scheme (e) monotonously rises till the solute concentration ratio in the raffinate and strip phases approaches a certain limiting (equilibrium) value equal to *m*_1_/*m*_2_.

When the equilibrium distribution is not attained in the cells and each cell represents an ideally mixed vessel, the process efficiency for scheme (Figure 5e) can be determined as follows:$$\frac{x_{1,N}}{x_{1,0}}=\frac{1-F_{1}F_{2}}{{\left[\frac{1+AF_{1}}{(A+F_{2})F_{1}}\right]}^{N}-F_{1}F_{2}},\;A=1+1/t_{2}+1/(F_{1}t_{1}),$$
where $t_{1}=a_{1}k_{1}V_{1}/v_{1}$ and $t_{2}=a_{2}k_{2}V_{2}/w$ are the numbers of transfer units in the extraction and stripping well-mixed cells, respectively; *V*_1_ and *V*_2_ are the volumes of cells; and *v*_1_, *v*_2_, and *w* are the volumetric flow rates.

To compare theoretical models with practice, experimental research was performed on a cascade of eight cylindrical mixer-settlers (mixer volume 50 mL, settler volume 400 mL) using the standard water–acetone–toluene extraction system [32]. Two series of experiments were carried out: 1. Staged version of supported liquid membranes with countercurrent flow of both mobile phases (scheme (e)): the membrane phase (toluene) recycles between each pair of extraction and stripping mixer-settlers. 2. Staged version of conjugated extraction stripping (scheme (h)): the membrane phase first passes through all the mixer-settlers of extraction, then through all apparatuses of stripping. During the experiments, the flow rate of the feed (*v*_1_) and stripping (*v*_2_) aqueous phases, as well as the circulating membrane phase (*w*), was measured, and samples of the aqueous phases were taken from each mixer-settler. The equilibrium distribution coefficients (*m*_1_ = *m*_2_ = 0.73) were determined in shake experiments. The results of the experiments, as well as the calculated theoretical concentrations, are given in Table 1 and Table 2.

The outlet concentration in the feed phase (x_1,n_) was calculated by Equations (22) and (25). The concentrations in all stripping and extraction stages were determined from material balance equations. The results in Table 1 and Table 2 demonstrate, in general, a satisfactory agreement between the prediction of mathematical modeling and the experiment, and confirm the advantages of the staged version of supported liquid membranes over the conventional extraction-stripping separation.

Compared to conventional conjugated extraction-stripping separation, liquid membrane separations have a number of advantages, especially when processing dilute solutions. Key benefits such as significant savings in reagents and solvents and better separation of components allow valuable compounds to be isolated and concentrated from dilute aqueous solutions, in particular, in wastewater treatment. The main advantage of solvent extraction methods over liquid membrane methods is the high throughput of modern extraction apparatuses such as mixer-settlers and multi-stage columns. The staged variants of liquid membrane extraction described above, called three- and multi-phase extraction [29,30,32,33], can be considered as a simple and effective way of practical implementation of the liquid membrane separation methods in industry. As mentioned above, this technique can be realized either in a series of mixer-settlers or in two- or multi-chamber extraction columns.

## 4. Conclusions

This review article is based on theoretical and experimental studies carried out by the authors and presents a mathematical description and comparative analysis of solvent extraction and liquid membrane separation processes with different flow regimes of contacting liquid phases. Equations are given for the simulation and designing of continuous and staged variants of the separation processes under consideration. A comparison of the processes of traditional extraction-stripping and liquid membrane separations showed that, from the point of view of the efficiency of the extraction, emulsion and film pertraction liquid membrane methods have advantages over the traditional extraction-stripping method, when the efficiency of the mass transfer at the extraction stage is much higher than the efficiency of the mass transfer at the stripping stage. The supported liquid membrane has advantages over traditional extraction stripping when mass-transfer rates on the extraction and stripping sides are different, while when they are equal, both processes demonstrate the same results. However, from a practical point of view, liquid membrane methods have significant advantages over solvent extraction, such as the low consumption of reagents and higher separation efficiency under certain conditions. In addition, because organic solvents are not removed from the process unit, liquid membrane methods are more environmentally friendly. The low throughput and complexity of liquid membrane methods can be overcome by using staged versions of these processes based on modified solvent extraction equipment (mixer-settlers or extraction columns). Experimental research performed on a cascade of mixer-settlers have demonstrated a satisfactory agreement between the prediction of mathematical modeling and the results of the experiment, and confirmed the advantages of the staged version of supported liquid membranes over the conventional extraction-stripping separation.

## Figures and Tables

**Figure 1 membranes-13-00554-f001:**
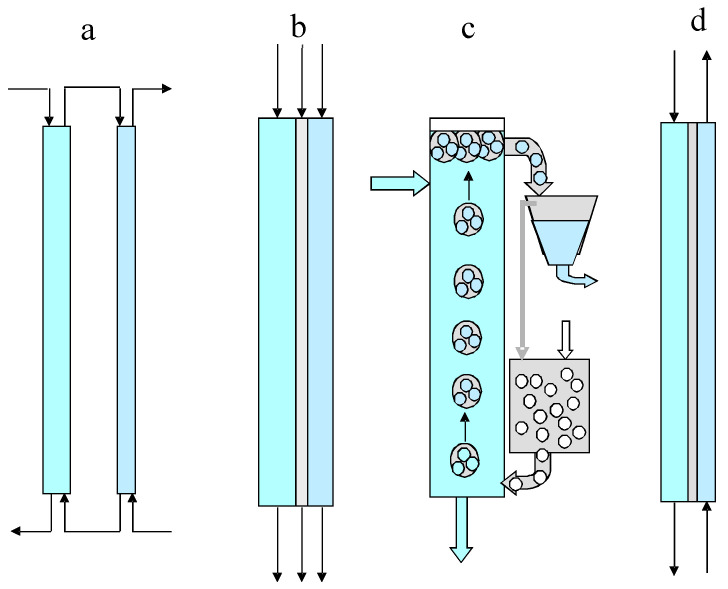
Schematic diagrams of extraction-stripping separation processes: a—conventional conjugated extraction-stripping; b—film pertraction; c—emulsion membranes; and d—supported liquid membranes.

**Figure 2 membranes-13-00554-f002:**
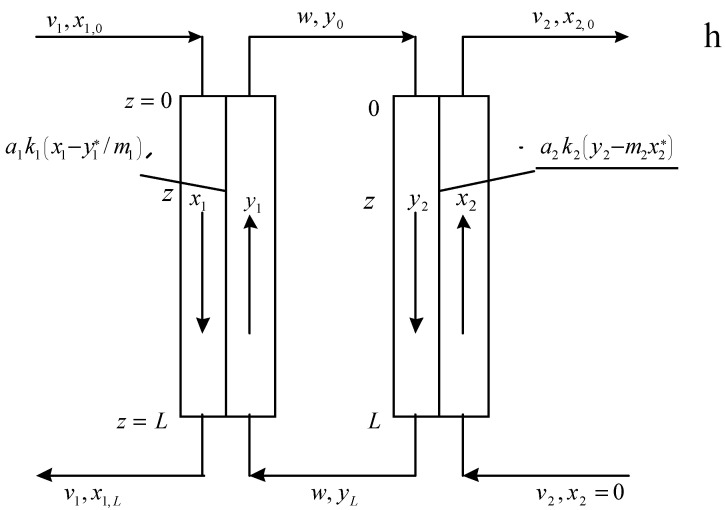
Flowsheet of mass transfer in a conventional conjugated extraction-stripping process.

**Figure 3 membranes-13-00554-f003:**
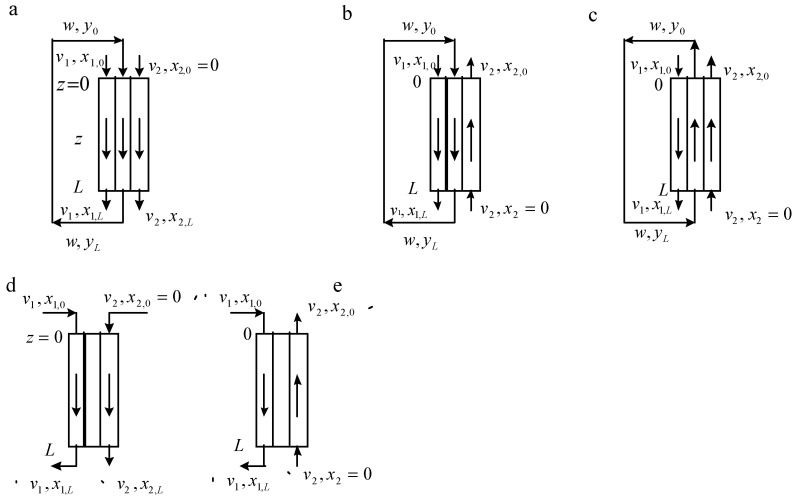
Flowsheets of mass transfer in continuous versions of liquid membrane technique.

**Figure 4 membranes-13-00554-f004:**
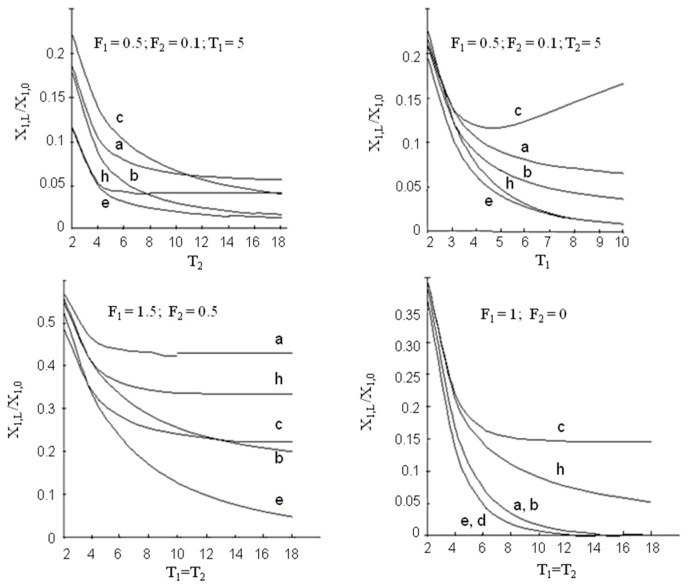
Comparison between efficiency of continuous schemes of extraction-stripping separation technique.

**Figure 5 membranes-13-00554-f005:**
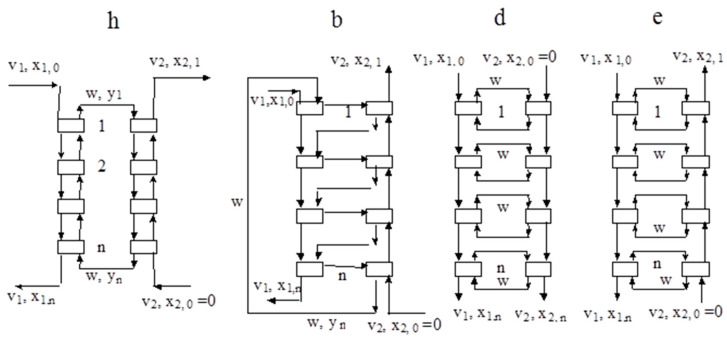
Staged versions of mass transfer in the three-phase liquid system.

**Figure 6 membranes-13-00554-f006:**
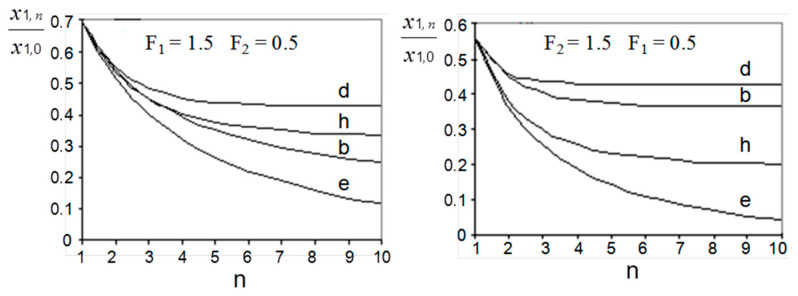
Comparison between efficiency of staged schemes of liquid extraction technique.

**Figure 7 membranes-13-00554-f007:**
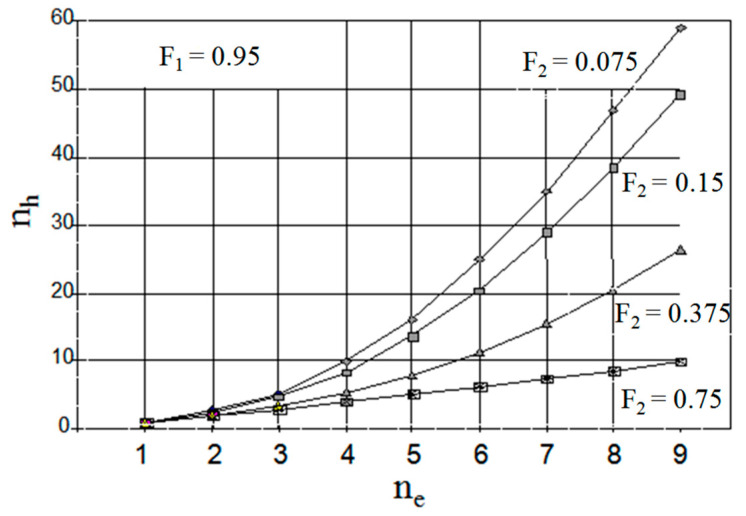
Comparison of necessary stage numbers in schemes (h) and (e).

**Table 1 membranes-13-00554-t001:** Theoretical and experimental concentrations in the mixer-settlers operating in the supported liquid membrane mode (scheme (e)).

Number of Stages	x_1_ (%) Experiment	x_1_ (%) Theory	x_2_ (%) Experiment	x_2_ (%) Theory
*v*_1_ = 1.12 L/h, *x*_1,0_ = 4.8%, *v*_2_ = 2.36 L/h, *w* = 3.0 L/h				
1	3.49	3.00	2.00	2.04
2	2.20	1.78	1.36	1.18
3	1.27	1.01	0.72	0.61
4	0.70	0.50	0.26	0.24
*v*_1_ = 1.48 L/h, x_1,0_ = 4.7%, *v*_2_ = 2.64 L/h, *w* = 6.0 L/h				
1	2.94	2.96	2.44	2.37
2	1.83	1.79	1.40	1.40
3	0.98	1.00	0.74	0.74
4	0.49	0.47	0.23	0.30
*v*_1_ = 1.15 L/h, x_1,0_ = 4.8%, *v*_2_ = 2.16 L/h, *w* = 10.0 L/h				
1	2.77	2.72	2.34	2.40
2	1.56	1.48	1.28	0.29
3	0.76	0.74	0.62	0.63
4	0.34	0.30	0.23	0.23

**Table 2 membranes-13-00554-t002:** Theoretical and experimental concentrations in the mixer-settlers operating in the conventional extraction-stripping mode (scheme (Figure 5h)).

Number of Stages	v_1_ = 1.15 L/h, x_1,0_ = 4.8%, v_2_ = 2.16 L/h, w = 10.0 L/h			
	x_1_ (%) Experiment	x_1_ (%) Theory	x_2_ (%) Experiment	x_2_ (%) Theory
1	1.84	1.86	1.83	1.85
2	1.43	1.40	1.81	1.82
3	1.37	1.33	1.71	1.71
4	1.36	1.32	1.40	1.32

## Data Availability

Not applicable.

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
