# Peer review of "Modelling and Comparative Analysis of Different Methods of Liquid Membrane Separations"

_membranes, 2023, doi:10.3390/membranes13060554_

Round 1

Reviewer 1 Report

This article proposes modeling methods for different liquid membrane separation methods. Propose and compare mathematical models for liquid membrane separation under different contact liquid phase flow modes. The results show that when the mass transfer efficiency of the extraction section is significantly higher than that of the extraction section, the liquid membrane method has advantages over the traditional conjugate extraction stripping method. Using improved solvent extraction equipment for liquid membrane separation can overcome the main drawbacks of liquid membrane separation: low flux and complexity. Considering the importance of this study, the revised manuscript may be worth publishing.

  (1)     There is a problem with the format of the reference document. The serial number does not use brackets.

 (2)     Can the author use other technologies to further support the morphology of NF film and composition of scale layer formed on its surface.

 (3)     If conditions permit, we hope that the author can provide a more detailed description of the specific modeling process.

 (4)     Related works on membrane separations are missing. (ChemPlusChem, 2023, 2023, 88, e202200423; Chem. Commun., 2023, 59, 1907).

Reviewer 2 Report

BRIEF SUMMARY

Interesting work, I have only a few remarks/corrections, which I present below. If authors improve/answer them, I could give the final “green light” for publication of this work in “Membranes” journal.

SPECIFIC COMMENTS

1.  Line 9Write “three-phase” and not only “three-”.

2.  Line 28As the letters of the symbols are all capital, begin each word with capital, also write “Solvent Extraction (SE)” and “Liquid Membrane (LM)”.

3. Figure 3: Each Figure must be appeared immediately after its first reference in the text. Therefore, transfer Figure 3 immediately after Figure 2, in Line 71.

4. Equations: Did you develop these equations? If no, you must give references for each one. You should explain each parameter after each equation, by giving also the units, f.e. Length, in [m].

5. Figure 4: Each Figure, with tis title/explanation, should be included in the same page. In this case, transfer Lines 170-171 in previous page 6.

6.  Lines 201-202: Transfer these Lines in next page 8.

7.  Figure 7: Similarly with comment No. 3, transfer Figure 7 after Line 231.

8.  CONCLUSIONSThis paragraph must be extended. The authors here should point the original parts of their work and their contribution in the existing literature. Why is this work important? Which are the new results/conclusion?

9. REFERENCES: Please check that all the references (names of authors and journals, titles, numbers of volumes and pages) are correct. Also check if you have followed the instructions of the journal about how to write the references and ensure that all the references are part of this work.

I would like to check it one more time before the final publication.

Minor editing of English language required

Reviewer 3 Report

This paper presents mathematical description and comparative analysis of solvent extraction and liquid membrane separation processes with different flow regimes of contacting liquid phases. They pointed out that the main disadvantages of liquid-membrane methods - low throughput and complexity - can be overcome by using modified solvent extraction equipment to carry out liquid-membrane separation. The manuscript can be further considered after revision. 

1. Authors listed their models and equations section 2. However, only simple descriptions about their results were provided. A discussion paragraph was presented in the end of Section 2, and only three reported ref. were compared. It is more like an experimental report rather than a research paper.  

Reviewer 4 Report

1. The comparisons are made under several assumptions. Please also discuss the limitations of the comparison that comes with the assumptions. 

2. please discuss potential applications of this review. 

The article is easy to understand. Please edit through the article on minor gramma issues. 

Round 2

Reviewer 3 Report

Accept as it is.